# Assessment of the Suitability of Selected Linear Actuators for the Implementation of the Load-Adaptive Biological Principle of Redundant Motion Generation

**DOI:** 10.3390/biomimetics9040236

**Published:** 2024-04-16

**Authors:** Marcel Bartz, Michael Jüttner, Fabian Halmos, Elias Uhlich, Max Klein, Patricia Drumm, Erkan Dreßler, Sina Martin, Jonas Walter, Jörg Franke, Sandro Wartzack

**Affiliations:** 1Department of Mechanical Engineering, Friedrich-Alexander-Universität Erlangen-Nürnberg (FAU), Engineering Design, Martensstraße 9, 91058 Erlangen, Germanymaxklein97@web.de (M.K.);; 2Department of Mechanical Engineering, Friedrich-Alexander-Universität Erlangen-Nürnberg (FAU), Institute for Factory Automation and Production Systems, Egerlandstraße 7, 91058 Erlangen, Germanyjonas.walter@faps.fau.de (J.W.);

**Keywords:** load-adaptive systems, musculoskeletal lightweight design, lightweight design robotics, linear actuators, redundant motion generation, tension chording, cable-driven robotics, rope pulls, artificial muscles

## Abstract

The load-adaptive behavior of the muscles in the human musculoskeletal system offers great potential for minimizing resource and energy requirements in many technical systems, especially in drive technology and robotics. However, the lack of knowledge about suitable technical linear actuators that can reproduce the load-adaptive behavior of biological muscles in technology is a major reason for the lack of successful implementation of this biological principle. In this paper, therefore, the different types of linear actuators are investigated. The focus is particularly on artificial muscles and rope pulls. The study is based on literature, on the one hand, and on two physical demonstrators in the form of articulated robots, on the other hand. The studies show that ropes are currently the best way to imitate the load-adaptive behavior of the biological model in technology. This is especially illustrated in the context of this paper by the discussion of different advantages and disadvantages of the technical linear actuators, where ropes, among other things, have a good mechanical and control behavior, which is very advantageous for use in an adaptive system. Finally, the next steps for future research are outlined to conclude how ropes can be used as linear actuators to transfer load-adaptive lightweight design into technical applications.

## 1. Introduction

### 1.1. Motivation and State of the Art

The human musculoskeletal system is a valuable source of inspiration for principles that can be used in engineering, especially in lightweight design [1]. The human extremities, which can be described mechanically as an open kinematic chain [2], have similarities to the kinematics of many technical systems, such as industrial robots, drive components such as coupling and crank shaft gears, and applications in mobile robotics and human–robot interaction [3]. Due to these similarities, the lightweight design of the musculoskeletal system offers great potential for increasing the load-bearing capacity and minimizing the use of resources in such technical systems.

Like any biological system, the human musculoskeletal system—the bones, muscles, ligaments, and tendons—can adapt to the changing environmental conditions [4,5], such as changing loads and motions. Two principles, in particular, have been identified as the mechanisms of adaptation: Firstly, in the long term, the adaptive remodeling of bone mass depending on the external load along the main tension paths (functional adaptation [6,7,8]). Secondly, in the short term, the kinematically overdetermined generation of movement of the extremities through a rapid and adapted interaction of the muscles, which is called tension chording according to Pauwels [9].

The principle of functional remodeling of bone mass is already well understood from a biological point of view [7,8,10] and is already used in technical methods, e.g., for the computer-aided structural optimization of technical components [11]. However, such optimizations can generally only be carried out once during the initial design but can hardly be adapted during operation. Therefore, for a load-adaptive operation, from a technical point of view, the imitation of functional remodeling of bone mass is considered to have less potential than the principle of tension chording [12]. Therefore, the principle of functional remodeling of bone mass is deliberately excluded from further considerations within this paper, while the principle of tension chording is furthermore considered in isolation.

Being the core aspect of the present study, the biological background of the tension chording principle with redundant motion generation (subsequently also referred to in this paper as the redundancy principle) is explained according to the current state-of-research. As an example, the flexion of the human arm is used, as shown in Figure 1.

The flexion of the human arm is generated by a greater number of muscles than kinematically necessary (as simplified with biceps and triceps, the brachioradialis and brachialis, and the deltoideus in Figure 1a). From a technical point of view, the following two theoretical (non-physiological) cases for generating movement would be possible. Although they do not correspond to the biologically correct flexion of the arm, they demonstrate well the principle of tension chording. In case 1, shown in Figure 1b, the arm is flexed by the action of the deltoid and the brachioradialis. The bending moment distribution, which is also shown in blue, indicates that the resulting bending moments in the upper arm are maximum, while they are minimum in the lower arm. In case 2, see Figure 1c, the deltoideus and the biceps act in contrast, whereby the circumstances of the bending are reversed, as can be deduced once again from the bending moment distribution. Thus, the bending moments in the upper arm become minimal, while they become maximal in the lower arm. Any other combination of muscles and also the strength of the forces can in turn produce different bending moment distributions along the upper and the lower arm. According to the tension chording principle of Pauwels [9], the human sensorimotor system finds an adapted muscular interaction, which allows an overall minimization of bending stresses [14] and, thus, homogenizes the stress in the bone spatially and temporally, saving mass and resources [5,6,7,9,15,16,17,18]. This optimal interaction was proven among other things by electromyography (EMG) measurements [19].

While being supported by the state of research, many principles of action are not yet sufficiently understood on a biological level to make the redundancy principle with tension chording applicable to technology. This is particularly true in a bottom-up process driven by biological push, in which a solution found in biology is to be used as a basis for a possible technical implementation. Reasons for this include the high complexity of the interaction of different principles and the inaccessibility of some measured variables. For example, muscle forces cannot be measured directly, and only activity patterns can be measured, as in Lutz et al. [19,20]. Though biomechanical simulation programs generate plausible motion data with energy- and bending-optimized algorithms [21,22], this is a major hurdle for an adequate understanding of the redundant motion generation in the musculoskeletal system. Therefore, the exploitation of the potential of tension chording is complicated by the incomplete knowledge of the mechanisms of action and the principles of the musculoskeletal system.

As a result, the principle of kinematically overdetermined motion generation for the load-adaptive minimization of bending stress through muscle forces has not yet been fully achieved in technical applications. Obviously, technical systems, such as cranes and masts, use tension cables as chords to brace the supporting structure, similar to the way muscles “brace” bones, in order to minimize bending moments and achieve normal stress in the components. However, these applications use a static principle, and the tension cables used as chords do not act as actuators there, like muscles do, but as passive elements. As an actuator, the tension chording principle is used in a simple form in some dynamic designs, such as biomimetic robots [23]. In many cases, passive and active tension chording elements are used for bending-minimized motion generation and damping [24,25,26,27,28,29,30]. However, in contrast to the biological model, the potential of the principle of the tension chording is usually not fully utilized because the motion generation with a redundant number of muscles is not applied. The few applications that use the principle of redundant actuators are similar to the biological model and mainly limited to biological demonstrators [23].

For the reasons mentioned above, there is a need for research to make the principle of motion generation with redundant actuators with tension chording available for technology. Therefore, the purpose of this paper is to contribute to the transfer of the principle of minimizing bending moments with redundant actuators to technical application. 

In addition to the specific biological uncertainties and the resulting need for research, there are definitely many issues that must be addressed on a technical side as well. The redundancy principle requires that the technical system must have overdetermined kinematics and actuators that can mimic the kinetics of real muscles. This is often not the case with the common use of rotational motors in technical systems, as these systems usually act as a link in the kinematic chain and therefore do not allow the kinematic overdetermination of the system. For this reason, this paper focuses on the technical requirements that can be placed on linear actuators in order to achieve redundant and bending-minimized motion generation based on the model of the musculoskeletal system in technical applications.

### 1.2. Research Objective of This Paper

The research objective of this paper is to assess the suitability of technical linear actuators for their use in redundant motion generation. The use of redundant linear actuators in technical systems results in a number of challenges. To cope with these, the first step is to select appropriate linear actuators, for example, rope pulls, pneumatic or hydraulic cylinders, or artificial muscles. In addition to the choice of actuators, there are further challenges, such as how many actuators to choose, which directly affects the degree of overdetermination of the motion generation, and at what positions and how the actuators should be fixed. Depending on the choice of actuator, other following issues also arise with respect to the dynamic behavior, and the stiffness as well as the response and control behavior of the actuators and the system.

This paper focuses on the first of these challenges, namely, the multifaceted choice of suitable actuators. Therefore, the possible actuator types for implementing the tension chording principle are assessed. Of course, some aspects of dynamic behavior and other issues are also considered in parts. In order to investigate the suitability of certain actuators under isolated boundary conditions, the actuators are first examined individually, i.e., in a non-kinematically overdetermined application. This means that the suitability for simple tension chording is checked here, whereby a future extension to redundant tension chording will be easy to achieve.

## 2. Materials and Methods

In the following, the biomimetic approach chosen in this paper is described first, and based on this, the technical model consideration and the methodology for assessment are presented.

### 2.1. Choice of the Biomimetic Approach

Since the load-adaptive behavior of the redundant interaction of the muscles in the musculoskeletal system is not yet fully understood, a biomimetic bottom-up/biological push approach would be suitable in this case, starting with a detailed analysis of the biological system [31,32]. However, due to the inaccessibility mentioned above, such as the inability to measure muscle forces and the difficulty in quantifying the interaction of muscles, a detailed analysis of the biological mechanisms of action is very difficult and, in some cases, not practical for a “simple” technical transfer. From a technological point of view, i.e., within the framework of a technology pull/top-down approach, when looking for analogies and a suitable example in nature, a similarity can be established between many technological systems and the musculoskeletal system. However, according to the analysis of analogies, it is not possible to refer to a “ready-to-use” described biological solution that can be applied in the technical system, since this does not yet exist in detail.

For the reasons mentioned above, a bioinspired approach, as described in detail in [33], is appropriate here. In this bioinspired approach, the current state of research concerning the biological effects is considered sufficient and is not further investigated on the biological side. Instead, the given state of knowledge, whether “right or wrong”, the known elements (such as muscles as actuators and joints and bones as structural components), and the mechanisms of action of the biological system are directly transferred—as known—to a technical system. The missing knowledge, especially about causal interactions, can then be obtained directly from the technical system. The advantage of this is that it provides an understanding of the “inaccessibilities” because the technical system can be studied both qualitatively and quantitatively, unlike the biological system. Of course, the correlations observed in the technical system may no longer be consistent with the biological model. Though it does not matter whether they are right or wrong, as long as the expected benefit from the biological principles is achieved on the technical side. This bioinspired approach is used in this paper as a biomimetic method. The system elements used in this approach are presented below in the description of the modeling. A detailed description of this approach can be found in [12,34,35].

### 2.2. Model Consideration

This section presents the mechanical model, which is the basis for the analysis in this paper. From the biological model, the musculoskeletal system and the biological model elements, i.e., the bones and the biological joints connecting them, are taken as mechanically abstract, simplified technical beam structures, and idealized technical joints, respectively, see Figure 2. The resulting kinematic chain is kinematically similar to the extremities of the musculoskeletal system but may contain different numbers of beam segments and joints in different arrangements and combinations. In most technical systems with such kinematics, the beam segments would be moved in a technically simple way with articulated arm motors (e.g., electric motors) arranged in the joints, see Figure 2a. This type of motion generation is often used because it is clearly defined in kinematic terms, which allows for simple and precise designs and controls. However, bending moments are induced in the beam segments during motion by applying torsional moments directly to the joints. In the context of modeling using the bioinspired approach described above, see the model in Figure 2b. The muscles that generate the motion in the biological model are abstracted as technical linear actuators in the context of modeling. In contrast to biological muscles, several technical linear actuators can generate compressive as well as tensile forces. In the bioinspired model, the joint motors, as shown in Figure 2a, are replaced by a large number of linear motors (represented by the force arrows in the figure). Due to the large number of linear actuators, it is possible to implement the tension chording principle described above with redundant motion generation by means of kinematic overdetermination. This bioinspired model, as shown in Figure 2b, is used in this paper as the basis for the further investigation and assessment of technical linear actuators and for the examination using two different demonstrators.

The number of forces *F*_Zn,i_ in the bioinspired model can be arbitrary, i.e., a number of forces i for each segment n. In contrast to the biological model, those forces can, but do not necessarily have to, be arranged antagonistically in a technical model. This means that with this initially rather generic model, many of the above-mentioned questions and challenges can be investigated. With our focus on the assessment of suitable linear actuators, excluding the issues of the position and number of actuators, only one pair of forces is considered per arm segment of the demonstrators, generating a kinematic determination. In contrast to joint motors, this still allows for the possibility of tension chording, which is made possible by the bracing.

With regard to the model presented here, it is important to emphasize that unlike the biological model, which is completely elastic, this proposed abstract technical model has only limited elasticity. The influence of stiffness and elasticity is not yet considered in this study. In addition, the model in this paper is a two-dimensional one, so that more complex relationships resulting from three-dimensional models are not yet overlaid upon the investigations.

### 2.3. Selected Methodological Procedure for the Assessment of the Linear Actuators

In order to assess the suitability of linear actuators for the implementation of the biological lightweight design principle of tension chording with redundant motion generation and, in the long run, to apply this principle in technical applications, in this paper, a twofold methodological approach is chosen to evaluate the suitability of technical linear actuators.

1.Literature-based methodology: Initially, the state of the art in the literature is purposefully summarized to determine which linear actuators exist. Since there is already a large number of technical linear actuators, partly independent of and partly inspired by biological models, this state of the art is first of all examined against the background of the possible use for tension chording. These results are then assessed against technical requirements. It should be noted that the results of this literature-based research are not all-encompassing, as this is not possible within the scope of this paper and is also not target-oriented. A more detailed explanation for this way of proceeding can be found at the beginning of Section 3, and the results of the first step are in Section 3.1.2.Experimental methodology: Based on the state of the art in the literature and the assessment of the suitability of linear actuators in the first step, two suitable actuators are selected in this paper as examples, and their performance is separately investigated and assessed using two individual simple technical demonstrators. The results of this step are presented in Section 3.2.

The results of the theoretical and experimental steps can then be combined, and the advantages and disadvantages regarding the possible challenges for real technical systems, all of which have different requirements, can be discussed and assessed by means of the proof of concept on the two demonstrators. This is carried out in the Discussion Section, i.e., Section 4.

## 3. Results

This section first summarizes the state of the art of linear actuators already available in research and technology and, based on this, assesses the suitability of these actuators for their use in the implementation of tension chording in technical systems. It is important to emphasize that the actuators and studies considered in this paper are not comprehensive in terms of the state of the art in research and technology, i.e., the aim of the paper cannot be a holistic review or a meta-study of the state of the art in linear actuator research. This means that it is restricted to “common” linear actuators and linear drives, such as those in standard mechanical and electrical engineering classifications as in [36] (in this paper, both mechanical linear drives and electrical linear actuators are subsumed under the term linear actuators). A possible selection here would be as follows:Artificial muscles;Rope pull actuators;Hydraulic actuators;Mechanical linear actuators, such as threaded rods;Electromechanical actuators, such as linear motors;Further varieties (no claim to completeness).

Furthermore, it is not the aim of this paper to review all research studies and all commercial products available worldwide. On the one hand, this would be a rather high level of effort at such an early stage, and, on the other hand, the aim of this paper is, first of all, to provide a general purposeful selection of possible linear actuator types and not a specific selection with special characteristics, since the types already differ widely from each other in their characteristics and the requirements that can be placed on them.

Therefore, in the following section, selected papers and reviews are presented, from which the basic properties of the different linear actuators can be derived. This does not mean that this research is fully exhaustive, and that conclusions drawn about the characteristics can always be generalized and applied to the entire state of the art in research and technology worldwide. This consideration is made in the context of this paper, because it is a first “rough screening” of the state of the art with the aim to quickly select two actuator types to examine them as soon as possible in a first demonstrator to generate a knowledge gain (see Section 3.2). Of course, identifying suitable actuators is an iterative process between reviewing the literature/state of the art and testing in demonstrators and also depends on the requirements set for the technical system.

### 3.1. Results of the Summary of the State of the Art for Linear Actuators

For the purposes of this paper, the purposeful selection of actuators discussed here is further restricted to the first two linear actuators from the bulleted list above. There are several reasons for this further limitation. Primarily, as explained in the introduction to Section 3, a comprehensive treatment of all actuators would go beyond the scope of this paper. Though the potential of the other actuators should not be neglected, it is reserved for a future study. Another reason for the preselection is their proximity to the biological model. They are often used as examples in biomechanics [23]. This is not the least due to the fact that, compared to hydraulic cylinders, they have a low weight, a small design space, and a similar stiffness to real muscles.

In terms of the bioinspired approach, we therefore remain relatively close to the biological system, with the state of the art of artificial muscles and rope pull actuators being discussed in Section 3.1.1 and Section 3.1.2, respectively. The explanation of the rope pull actuator is more detailed than that of the artificial muscles. This is because artificial muscles have not yet been widely used in technical systems. Rope pulls, on the other hand, have been in use for a long time, but in view of the biological principle of redundant tension chords, the surrounding infrastructure, i.e., deflections, drives, etc., should also be examined more closely.

#### 3.1.1. Results to the Summary of the State of the Art of Artificial Muscles

To date, many different types of actuators have been developed for use as artificial muscles. Within this paper, the term artificial muscle is used to refer to a class of actuators that can reversibly contract, expand, or rotate within a single component in response to an external stimulus [37]. This means that they do not need an additional drive. The actuators differ significantly in their operating principle and, consequently, in their structure, mass, dynamics, and thus in their application. In order to cover as wide a spectrum as possible, four actuators are discussed below that are considered to be artificial muscles [37,38,39,40] and are based on completely different operating principles: pneumatic artificial muscles, dielectric elastomers, shape memory alloys, and yarn artificial muscles.

Pneumatic artificial muscles (PAM), also known as rubbertuators or braided pneumatic actuators, usually consist of a thin, flexible, airtight membrane that forms a pneumatic bladder and a braided mesh sleeve. The mesh sleeve may be integrated into the membrane or wrapped around it. When compressed air is applied to the artificial muscle, it can deform in different ways, such as contracting or expanding, depending on the design of the braided mesh sleeve [40]. PAMs can be divided into several types that have been developed over the last decades. For details and a list of a large number of application examples, see a recent comprehensive review by KALITA et al. [38]. To provide a few examples of bioinspired soft robots, the elephant trunk and the octopus arm are often used as models [41,42,43,44,45]. However, PAMs are also used in medical applications for the human–robot interaction, where conventional electric drives can be dangerous due to the high stiffness of the joints and the high weight of the electric motors [38]. Common to all types of PAM is the advantage that they resemble biological muscle, especially the contractile variants, through linear contraction with a monotonically decreasing load–contraction relationship [38]. 

However, in addition to the specific advantage mentioned, there are also a number of disadvantages that stand in the way of their more widespread use. For example, the control of technical systems with different artificial muscles is difficult because external loads can change the shape of the muscles themselves [42]. This is particularly important when it comes to controlling the already complex overdetermined tension chording. Furthermore, although the artificial muscles themselves are very light, the compressors that supply the compressed air are quite large and heavy, making it difficult to miniaturize systems using PAMs. This could limit the application for the implementation of the above-mentioned tension chording principle for active bending minimization with regard to precise control.

However, dielectric elastomers (DEAs), belonging to the group of electroactive polymers, are much smaller and use a completely different, electrostatic principle. An incompressible polymer film is placed between two flexible electrodes, similar to a plate capacitor. When an electric charge is added to the electrodes, they charge in opposite polarities, creating an electric field. This causes an attraction of the electrodes, compressing the polymer film and causing an expansion perpendicular to the applied force due to volume constancy [46]. To increase the forces that can be generated by a larger surface area, but by keeping the installation space small, the DEAs can be rolled up tightly, separated by an additional layer of insulation. The result is a cylindrical actuator that is electrically contacted at both ends [47]. DEAs allow quite a large strain with a fast reaction time. Taking into account the additional criteria such as actuation pressure, density, and efficiency, PELRINE et al. [39] considered DEAs to be the closest to biological muscle in animals. However, one obvious difference is that the active movement of DEAs is a relaxation, not a contraction, like biological muscles. In addition, the low stiffness of DEAs prevents them from generating compressive forces during expansion, as they quickly buckle. This behavior that is opposite to biological muscles requires a rethink when transferring biological models to technical systems. In order to generate movement, the muscles need to be pretensioned in the application, which is cancelled out by the expansion of the DEA, and if necessary, by the additional use of gravity. To control DEA, voltages in the kilovolt range are still required, which makes some applications more challenging. Scalability is also difficult and tends to restrict use to small dimensioned applications. Here, however, the very thin design can be of benefit, particularly in medical applications such as bionic dielectric elastomer iris actuators [48]. In addition, bioinspired soft robotics have also been developed with DEAs [49,50]. In principle, the kinematics of DEAs appear to be suitable for use in tension chording with bend minimization, but this requires a large number of actuators, which may be difficult to implement due to the high voltages and current flows.

Shape memory alloys (SMAs) are based on a thermo-mechanical principle. A special alloy, usually Nitinol (a nickel–titanium alloy), can be used to restore a previously defined shape after deformation by heating. This recovery is based on a reversible phase transformation from martensite to austenite. A resulting disadvantage is that the dynamics are limited because cooling is slow after the heat-induced recovery. SMAs are most commonly used in the wire form. The often cited disadvantage of a limited stroke can be mitigated by the clever arrangement of the wire [51,52]. For example, longer strokes with lower forces can be achieved by using pulleys to transmit the wire or by winding the wire into springs. SMAs are also found in robots inspired by the animal kingdom, such as the neck of an owl [51] or, again, the arm of an octopus [53]. From the kinematic point of view, SMAs are similar to the ropes discussed in the following section and, thus, also have a number of advantages in terms of use for tension chording. However, for active bending minimization, both a fast and a reliable control behavior is required, which SMAs do not have at the current state of the art.

Yarn artificial muscles (YAMs) consist of twisted fibers that can also be coiled and braided depending on the application. YAMs made of high strength polymer, such as nylon, react to heating by changing their length [54]. Unlike SMAs, they show hardly any hysteresis in stroke between heating and cooling [54]. In contrast, YMAs made of materials such as viscose or wool show a humidity-driven reaction, which means that they contract when exposed to moisture [55,56]. The advantages of YAMs are good availability and biocompatibility. In addition, the yarns are flexible so that they can, for example, be integrated into products in the textile industry with an aim of obtaining smart textiles [55]. For technical applications, their main disadvantage is the possibility of actuation. For example, the controlled moistening and drying of the YAMs are unnecessary challenges compared to other artificial muscles for the given application.

Despite their very different operating principles, the examples of artificial muscles presented share some common features that are worth mentioning. They are all easy to manufacture, which, together with readily available and cheap materials, makes them very cost-effective. Furthermore, they also have high power-to-weight ratios because they are lightweight but can generate large forces. With mostly fast reaction times and a certain damping effect, the characteristics of artificial muscles are close to those of biological muscles. This makes them well suited to soft robotics. Nevertheless, they have so far found little use beyond the mentioned bioinspired systems in the field of soft robotics [57] and medical technology. This is partly due to the actuator-specific drawbacks, and more generally due to their limited scalability and relatively complex control for movements over long distances and for applications requiring high accuracy and repeatability.

#### 3.1.2. Results of Ropes as Actuators

Ropes are very similar to muscles as actuators in the biological role model, i.e., in the musculoskeletal system, because like muscles they can only transfer tensile forces. However, unlike (artificial) muscles, they are not active actuators, so they are usually used as passive elements in rope pulls with an additional drive. Therefore, external motors are required to drive the ropes, which are often located away from the moving segments and joints (e.g., on the floor), but sometimes also on the moving components (e.g., on the joint or housing) [58]. The aforementioned differences from (artificial) muscles initially appear to be a disadvantage, as ropes always rely on additional infrastructure and aggregates such as deflection and guide pulleys. However, the use of ropes also has advantages, as they have, for example, a simpler design than artificial muscles, and as the load-bearing structure of the actuator and the drive are functionally clearly separated, and therefore, the ropes have well-matched properties depending on the material, and these properties can be adjusted. Another advantage is the simpler manufacturing, availability, and for some also the cost. The mentioned characteristics and other issues relevant for the use of ropes as actuators for the implementation of the tension chording of the human musculoskeletal system are explained in more detail in the following state-of-the-art.

Ropes are linear and flexible elements that are primarily used for the transmission of tensile forces and consist of metal, synthetic, or natural fibers that are twisted or interwoven with each other [59]. The guideline VDI 2500 [59] classifies ropes according to their design into twisted fiber ropes, hawser and lay ropes, braided fiber ropes, and round braided ropes. Another classification is made according to WEHKING [60], among others, according to which ropes can generally be divided into moving ropes, fixed/bracing ropes, suspension ropes, and sling ropes. Moving ropes are mainly used in elevators or for rope drums. Fixed ropes are used, for example, for bracing bridges. The suspension ropes are known from cableways, and the sling ropes are used, for example, in load loops for lifting loads [61].

For the implementation of the tension chording principle, it may initially seem reasonable to use standing ropes or suspension ropes, since bracing ropes, such as those used for bridges or cranes, are technically passive tension chords, as mentioned in Section 1, where steel ropes are often used. However, the tension chording principle in the human musculoskeletal system is not a static principle, resulting in the need for drives. As those are usually located outside the moving system or joints, additional deflection units must be provided. It has to be ensured that the ropes are bent with a radius that is significantly larger than the rope diameter [62]. Therefore, due to their low flexibility [61] and high moving mass, steel and other bracing ropes are not well suited for the implementation of the tension chording principle, along with natural fibers due to their weaker properties. For this reason, synthetic fibers present a suitable material for the implementation of the biological tension chording principle as moving ropes, which is considered further below.

Besides simple polyamide fibers, there are so-called high-modulus–high-tensity (HM-HT) fibers, which were developed using aramid. Modern HM-HT fibers use, among others, high-modulus polyethylene (HMPE) or synthetic high-performance polymers (PBOs). According to ROST [61], tensile strength and abrasion resistance are the most important properties for the selection of ropes due to their significant effects on the potential of a rope pull with synthetic fibers. Figure 3 shows the tensile strengths of different types of ropes.

Since HM-HT fibers have a higher tensile strength than natural fibers and standard polymer fibers, they offer a suitable choice, since they are not necessarily much more expensive than the other fibers.

As mentioned above, correct rope guidance and deflection play an important role [61], since the guiding mechanism is omnidirectional [62]. According to [61], there are several options for rope deflection, for example, with deflection rolls or channels with variable or constant bending radius. Here, the deflection rolls with rolling or journal bearings have significantly lower friction than, for example, channels. However, channels can be much more space efficient than the bearing-mounted deflection rolls. The friction in the channels can lead to non-linear friction effects that directly affect motion generation and transmission [64]. These effects have also been studied by PALLI and BORGHE-SAN et al. [65] and KANEKO and WADA et al. [66], where internal and external rope friction, friction of deflection bearings, and friction of seals also play a role [67]. In contrast, optimal friction is desired for deflection rolls so that the ropes do not slip. Due to the different advantages and disadvantages of rope deflection systems, it is necessary to decide which is most suitable depending on the application. In addition, there are mixed designs which combine sliding channels and bearing-mounted deflection pulleys [61]. 

From the characteristics of the deflection units described above, it appears that deflection rolls rather than channels are more suitable for the implementation of the tension chording principle. This is due to the fact that in the case of adaptive bending minimization, minor additional forces applied to the structure are desired. Friction in the channels can generate unwanted additional forces and the resultant bending in the structure, which must be compensated for. Furthermore, the higher rope control and transmission behavior of deflection rolls make them suitable for use in adaptive control systems that have to act rapidly.

As previously described, ropes as actuators differ from (artificial) muscles by being passive elements. This means that, like tendons as passive elements in the musculoskeletal system, they can only generate a force through pretension as a result of an external tensile load. This pretension is useful in biological systems for various reasons, e.g., as energy storage, but also for the tensile chording of the bones [9]. Therefore, it would be useful to include this in the transfer of tensile chording to the technical applications. Pretension is also described for technical ropes as an externally generated beneficial effect on the stress state [68]. The state of the art shows that there are various ways of achieving rope prestressing in service. According to ROST [61], a general distinction can be made between pretension with tensioning pulleys or rope tensioners. In both types, a fixed or variable tension is applied to the ropes in the rope path or at the end. The variable pretension is introduced passively by a damper element or actively by an actuator. In the case of tensioning pulleys, it should be noted that a high level of friction between the pulleys and the rope is good for the function.

For rope pretension devices, the pretension options for belt drives according to WITTEL and JANNASCH et al. can be used [69]. These include tensioning rails in which the motor can be moved by means of adjusting screws so that the rope has tension. Furthermore, tensioning slides should be mentioned, in which the tensioning is carried out automatically via weights and springs, which are attached to the mounted motor. Finally, tensioning rockers can also be used, in which the motor stands on a rocker and, depending on the direction of rotation, the reverse torque generates an automatic tension [69]. In summary, it is not possible to make any general statements about the advantages and disadvantages of the rope pretension options with regard to the biological tension chording principle, as this is highly dependent on the specific technical implementation and the requirements.

Another important issue for the implementation of the tension chording principle with redundant motion generation by means of ropes concerns the position of the drives of the rope pulls. They can be mounted outside the moving elements (e.g., on the floor, ceiling, or housing) (see Figure 4b) or can be attached along with the moving elements, e.g., in the joints (see Figure 4a, where the lower drive is attached to the floor). The first situation is advantageous in terms of the moving mass, which can be kept low. However, this requires many rope deflections and guides, which in turn introduce loads into the structure through, among other things, joint forces, which counteract the actual tension chording. Whilst it is often possible to avoid this by attaching motors to the moving elements, the increased moving masses introduce bending moments due to inertia, which is not desirable. Since both moving and static motors are not similar to the biological model, no general statement can be made here on the implementation of bending minimization by the tension chording principle in technical applications, and it must be carefully evaluated in which case the undesired bending moments can be tolerated better, which occur by the drive and the rope guide. Finally, it should be noted that the above-mentioned problems do not apply to shape memory alloys used as ropes [51], though they appear unsuitable for use in an adaptive system, particularly due to their high actuating times.

In addition to the arrangement of the rope drive, the achievement of positive and negative rotational movements of the segments has to be considered. In the human musculoskeletal system, there are agonistic and antagonistic muscles for this purpose. In technical rope-driven system, the rotational movement for both directions can be performed in a simple way with two motors and two ropes per segment to perform this motion in the agonist/antagonist principle. In order to use only one motor per degree of freedom in multi-segmented robots, it is necessary that the movements of the antagonistic and agonistic ropes are symmetrical [58], as they are in Figure 4b. In this case, one motor can be used with two ropes wound in opposite directions on the output shaft. Thus, when the motor rotates, the agonistic rope can wind up, and the antagonistic rope can unwind. However, no additional pretension can be achieved in this way, since one rope is pulled and the other rope is relieved. In the biological tension chording principle with redundant actuators, agonistic and antagonistic muscles often act simultaneously and adaptively compensate for external bending loads through pretension and active load application. Therefore, for the technical implementation of the tension chording principle with ropes, only the variant with two independent agonistic and antagonistic actuators is suitable, i.e., with one actuator per rope.

Finally, it should be mentioned that there are many recent studies on muscle and rope-like actuators such as the Verdrill muscle, which is also called the strand muscle Actuator [70]. In this type of muscle, two motors are required per degree of freedom, twisting two or more fiber ropes until they contract, thereby creating a traction force towards the motor [71]. These muscle-like drives also include the DoHelix muscle and the Quadhelix muscle, in which only one motor per degree of freedom is required [61]. In the DoHelix muscle, developed by STAAB et al. [72], a thin, highly flexible plastic fiber rope is wound onto a shaft attached to an electric motor. When the muscle contracts, the rope runs towards the shaft from two opposite directions and winds itself into a helix on the shaft. Further approaches of this kind can be found in the current state of research, but they are not yet available at the market level, and therefore, they are not presently suitable for implementing the tension chording principle in technical applications.

### 3.2. Results of the Demonstrators

From the literature summary in Section 3.1 and Section 3.2, it is clear that there are various properties as well as advantages and disadvantages for the use of artificial muscles and ropes for the implementation of the tension chording principle in technical applications. Some properties have been shown to be suitable for tension chording, while other properties have not been shown to be suitable, and some aspects are not yet fully conclusive and require further research. In order to generate more in-depth knowledge and to further investigate the results of the theoretical literature summary, two different technical demonstrators were built. Based on the results of Section 3.1, an artificial muscle was chosen as the actuator for one of the demonstrators and rope pulls for the other demonstrator. No redundancy is required to check the suitability of the linear actuators for use as tension chords. Therefore, both demonstrators are kinematically determined, which simplifies control and the general design. The results of the experimental part are discussed below. Due to their individual design, the demonstrators should be treated as proof of concept in separate cases. A head-to-head comparison is not intended.

#### 3.2.1. Demonstrator with Artificial Muscles

The literature summary on the state of the art in research in Section 3.1.1 has shown that there are several artificial muscles for different applications, but they are not yet used on a larger industrial scale. Currently, the use of artificial muscles is particularly suitable for applications in biotechnology and medical technology. As explained above, ropes appear to be more suitable as actuators for implementing the tension chording principle for technical applications. Although many approaches for more industrial applications are still in the research stage, this paper explores the potential of implementing the tensile chording principle with artificial muscles using a simple technical demonstrator in the form of an articulated arm robot.

For this demonstrator, DEAs are selected as artificial muscles, since these are available to the authors (i.e., those from the Institute for Factory Automation and Production Systems) [48,73] and are already being used for biomedical applications. As stated before, the main difference between DEAs and biological muscles is that they do not contract when “tensed”, thus introducing tensile forces into the structure. Instead, they expand, thus exerting compressive forces. This characteristic requires a complete reconsideration of the introduction and action of the forces on and within the support structure. Figure 5 shows the concept chosen in this paper for the investigation of the implementation on a simple two-articulated arm robot.

The kinematic design is based on the kinematics of the upper and lower arms shown in Figure 1. In contrast to the biological upper/lower arm system, the geometric dimensions in this demonstrator design are scaled proportionally (approximately 1:4, see dimensions in Figure 5a). Moreover, not all muscles acting on the arm are used, but only two muscles are used as actuators, which take over the functions of the biceps and the brachioradialis in analogy to Figure 1. The arm, which is bending minimized by the tension chording, is the beam labeled “arm” in Figure 5a. The “pillar” takes over the task of the articulated support of the arm and the mounting of the two actuators. Each of the DEAs are articulated and thus connected torque-free to the arm and the pillar. The demonstrator is additively manufactured almost entirely from the material PLA (polylactic acid). The arm as a beam is made of PC (polycarbonate) in order to make the influence of the tension chording visible with the use of photoelasticity, see Figure 6b. Figure 6a shows the physical demonstrator with a load *F* resulting from a mass of 25 g at the end effector of the arm and the resulting stresses in Figure 6c, visualized by the photoelasticity.

Even under static loading, i.e., without active actuator activation, the elastic DEAs act as tension chords, since they are elongated by the weight applied to the end effector, thus tensioning the arm by means of the elastic restoring forces. This tension chording can be seen due to the photoelasticity in Figure 6c. In Figure 6c, bluish color indicates compressive stress, and red-yellowish color tensile stress. It can be seen that a large area of the beam is under compressive stress, especially near the joint, where without tension chording high bending moments would occur due to the large lever arm of the external load. Tensile stresses prevail only near the applied external load and in the upper part of the beam, indicating some residual bending, especially in the part of the beam near the end effector. However, due to the lever arms, bending stresses at the end of the beam are expected to be lower than on the other side of the joint, so that the passive tension chording works through the elastic-acting DEAs and thus minimizes bending over a large fraction of the arm.

The beneficial bending stress behavior of artificial muscles, in general, and DEAs, here specifically, is advantageous, but it could also be achieved by other passive elements, such as springs. Although the DEAs show a good mass-to-elasticity ratio. The dynamic tension chording, i.e., the active motion of the arm supported by the muscle, is also interesting. However, in this setup, it has no additional effect on the bending minimization, since this is already given by the bracing and, as there are no redundant actuators, bending cannot be actively minimized. The main dynamic properties of DEAs are similar to those described in the state of the art. 

In summary, with the DEAs used here, it was possible to achieve tension chording with the sufficient minimization of bending stress. Due to the chosen longitudinal effect of those artificial muscles, i.e., the fact that they elongate and do not contract when activated, their use in the context of technical systems such as those described above is not sufficiently feasible. However, systems can also be redesigned from “pull actuators” to “push actuators” [74], such as those used in heavy construction machinery in the form of hydraulic cylinders. However, the DEAs used here do not bend stiffly and do buckle under a compressive load. Therefore, as actuators, they are currently more suitable for biomedical or biology-related applications, such as those in [48], instead of being used as a substitute for joint motors in “classical” technical systems by linear actuators in order to apply the tension chording principle by means of redundant actuators.

#### 3.2.2. Demonstrator with Ropes

The literature summary in Section 3.1.2 shows that the use of rope pulls as actuators seems to be a suitable choice to consider the tension chording principle on technical systems, even if joint forces and deflections have to be considered as possible negative side effects, and further technical efforts have to be made. Therefore, in this paper, the use of HM-HT ropes, which seemed to be a suitable choice based on the literature summary, is investigated on a two-articulated arm robot as a demonstrator. The design of this robot is shown in Figure 7 and roughly corresponds to the kinematics of the upper extremities of a human body. However, this demonstrator is abstracted so far that it rather corresponds to a classical industrial robot with two joints. The scaling of the geometric dimensions is approximately 1:2 to that of a human arm.

Since a kinematically determined actuator system is investigated in this paper, two rope pulls would be required for the two arm segments, and thus, requiring two degrees of freedom. Since ropes can only transfer tensile but not compressive forces, each arm segment also needs a rope on the opposite side to be able to move in positive and negative angular directions, which means that there are four rope pulls. This is analogous to the agonist/antagonist drive principle described in Section 3.1.2. The ropes are controlled by stepper motors, so that each individual “tension chording” can act individually. The final manufactured demonstrator is shown in Figure 8.

First, the static mechanical behavior of the demonstrator is described. In contrast to the artificial muscles considered above, the ropes used here have a higher stiffness; although a certain pretension is possible and thus also a passive tension chording, this cannot necessarily be used actively to passively minimize the bending in the arm segments, but rather to enable the precise movement of the arms without clearance. However, ropes have good damping properties, which are particularly important in the case of dynamic mechanical behavior. Passive bending minimization is definitely provided by the ropes in this demonstrator, as they act in a similar way to the static bracing mentioned above, as in the case of cranes.

The control of the ropes, i.e., the application of force, shows that this is more precise, faster, and more efficient from a control point of view, especially compared to the demonstrator with the artificial muscles. The accuracy of positioning, especially of the end effector, can be well ensured by the pretensioning of the ropes and by controlling the individual wire rope pulls using the stepper motors. Compared to industrial robots with articulated motors, this demonstrator definitely has a higher compliance due to the ropes, and there are also instable positions due to the kinematics (especially, since it is a kinematic chain with non-linear dynamics). This can certainly be an advantage for some robot systems in human–robot collaboration or in the context of soft robotics.

Finally, it should be mentioned that, unlike the demonstrator with the artificial muscles, no measurements of the bending stresses in the arm segments have yet been made for the rope-operated demonstrator. However, results are available for a similar system from a multi-body simulation, where it was shown that the bending, especially in the arm segment of the end effector, is minimized by the ropes acting as the actuator [12,35]. Nevertheless, a certain amount of bending stress remains, due to the fact that the ropes do not act directly from arm segment to arm segment, but have to be deflected via the joints, resulting in undesired bending moments due to the joint forces that occur there. However, our own preliminary research shows that these bending moments can be specifically minimized by redundant actuators, which is not within the scope of this paper [12,35].

## 4. Discussion

In the context of this paper, different artificial muscles and rope pulls, in particular, have emerged as the possible linear actuators for the implementation of the tension chording principle with redundant actuators on technical systems. Both in the literature summary and in the examinations with the demonstrators, the advantages and disadvantages of the characteristics of those different actuators have been determined in each case. The advantages and disadvantages are summarized in Table 1 and discussed below.

In general, artificial muscles kinematically resemble biological muscles. They can contract or expand on their own without any additional drive. This is particularly advantageous because they can be mounted directly between two segments, applying forces directly at the point of attachment without the need to divert forces. Power can also be supplied at the point of installation or simply connected. In addition, the elasticity of most models allows them to act as passive tension chords, similar to springs, so that they do not need to be active actuators at all and can still minimize bending moments.

On the other hand, many of the artificial muscles are still in the research stage or are mostly suitable for biomedical applications. Therefore, they are difficult to use in classical mechanical engineering, where they could replace joint motors, partly because of their dynamic behavior, which has not been studied in detail in this paper. This means that controlling them for precise positioning can become very challenging. As most artificial muscles are not yet available on the industrial market, they are often expensive and, therefore, not available in sufficient quantities. In some cases, additional equipment must be purchased to operate the muscles, such as high voltage sources for DEAs or pneumatic systems for PAMs. For many systems, especially simpler and mobile systems, this is uncommon and tends to act as an inhibiting factor. Finally, it should be recalled here that the scaling of artificial muscles is limited, which is particularly disadvantageous for technical applications beyond soft robotics and medical technology.

Ropes as linear actuators are still similar to the biological model. In any case, they also require additional equipment, mainly external drives and often even more cable deflections and guidance systems. However, these are cheap and easy to purchase in large quantities. A major disadvantage of rope pulls is that making direct connections from one segment to another is not possible. In most cases, the necessary deflections, joint forces, and other fastenings introduce unwanted additional loads into the structure, which can counteract the complete bending minimization, at least in the case of non-redundant actuators.

On the other hand, ropes are very versatile in terms of their mechanical properties due to their different material choices and very robust in terms of control and regulation as they have been used in a wide range of industrial applications for decades and are commercially available. Ropes can transmit high forces and often have a well-known linear–elastic behavior, which makes them attractive for accurate and precise control, as required especially for the tension chord principle with redundant actuators. Fast control also benefits from the fact that ropes have very low moving masses. This means that fast movements can be performed without large inertial forces that could prevent active bending minimization. Ropes have a lower stiffness and, therefore, have a different system behavior for drive systems compared to traditional joint motors. This lower stiffness can also be an advantage for systems where compliance is desired. Finally, the wide variety of rope materials, rope arrangement, and guidance, as well as the choice of drive type and drive position, results in a high degree of design flexibility to implement tension chording in different technical systems.

Due to the predominant advantages of ropes in the implementation of the tension chording principle, they appear suitable for a future study on the transfer of this lightweight design principle to technical applications. However, a number of aspects have not yet been investigated in this study, including a more detailed study of the dynamic behavior and control aspects. In addition, the use of the system within a redundant actuator system has not yet been the subject of a detailed investigation and will be the subject of a future study. This is a prerequisite for the implementation of a load-adaptive system using ropes as actuators. Conclusions for possible future study and other aspects are discussed in the following Section 5.

## 5. Conclusions

The aim of this study was to evaluate the suitability of technical linear actuators for the generation of redundant motion for use in a load-adaptive technical system with tension chording based on a biological model. The investigation was carried out by means of a purposeful literature summary and the study of two simple demonstrators. It was shown that there are many technical linear actuators that can be used to attempt to reproduce the redundant biological muscle interaction in order to act on structures in a load-adaptive manner. However, it became clear that there is no single actuator that combines all the desired properties, but with each actuator, there are certain trade-offs. Nevertheless, ropes have been found to be a particularly suitable actuator for implementing the tension chording principle in technical applications. For this reason, the following discussion will always refer to the studies of technical systems using ropes as actuators.

As described in the introduction of this paper, there are many challenges to be addressed in order to technically implement a successful transfer of the tension chording principle with redundant actuators. Some of these challenges have been addressed in this paper, while others have not yet been addressed or new questions have arisen through the study described in this paper. The first step has been taken towards the possible choice of actuators, but further research is needed to address the research questions and challenges that are described below:Detailed studies on dynamic system behavior, elasticity, and damping by using ropes as actuators.Detailed investigations into motion generation with redundant actuators and the associated advanced aspects, such as the number of rope pulls (the degree of kinematic indeterminacy), their positioning, and the necessary design rules. Additionally, the identification of the possible systems that can benefit from tension chording. Control engineering questions as well as path control related questions, which can also be used to actively minimize bending (which was not considered at all in this paper).Further research into artificial muscles and the possible marketability of suitable actuators, as this type of actuator may become relevant again.

In summary, this study has shown that, within the framework of the chosen research methodology, rope pulls currently appear to be the most suitable actuator technology for implementing the tension chording principle with redundant motion generation, and therefore, further research should be carried out on this type of actuator. The long-term goal is the implementation of the load-adaptive behavior in technical applications in order to save resources and energy in drive technology and robotic systems. In this way, their broader use in drive technology and robotics can be investigated as well as their real-world impact.

## Figures and Tables

**Figure 1 biomimetics-09-00236-f001:**
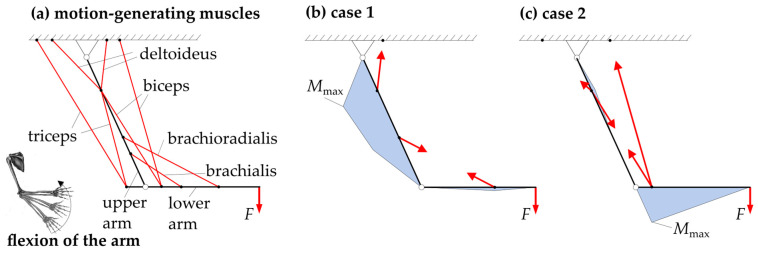
Demonstration of the principle of redundancy: the flexion of the human arm by the kinematic overdetermination of the motion-generating muscles according to [13]. The red lines are the labelled muscles and the red arrows are selected associated muscle forces.

**Figure 2 biomimetics-09-00236-f002:**
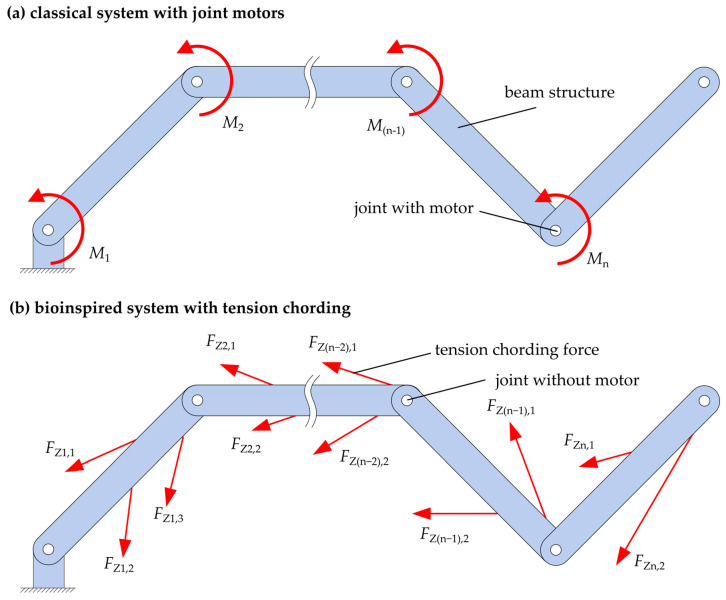
Kinetic model consideration of a kinematic chain in the case of motion generation with joint motors (**a**) and with redundant actuators with linear actuators (**b**). *M* are moments in the technical joint, and *F* are forces of the linear actuators. Index n denotes the segment (beam structures) according to [35].

**Figure 3 biomimetics-09-00236-f003:**
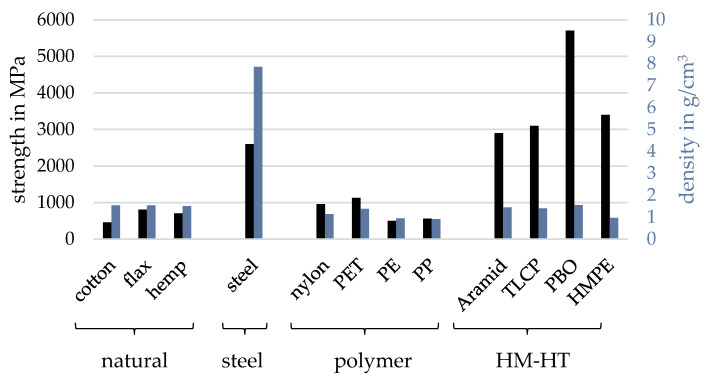
Strength and density of major rope fibers, based on [63].

**Figure 4 biomimetics-09-00236-f004:**
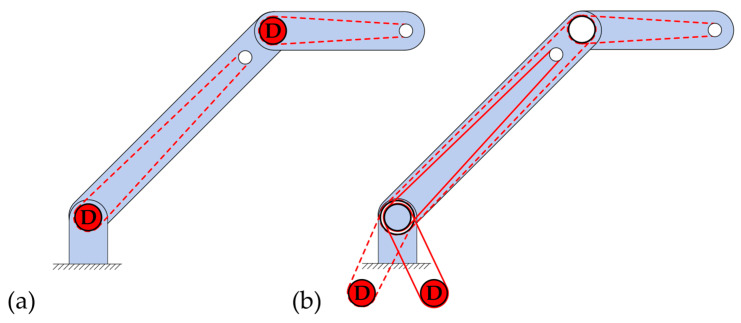
Options for the arrangement of the drives (Ds) of the rope pulls for the movement of the arm segment: joint drives (**a**) and drives mounted outside the moving elements (**b**). The rope pulls are shown in red. In order to be able to distinguish rope pulls that belong together, some of them are shown as dashed or solid lines.

**Figure 5 biomimetics-09-00236-f005:**
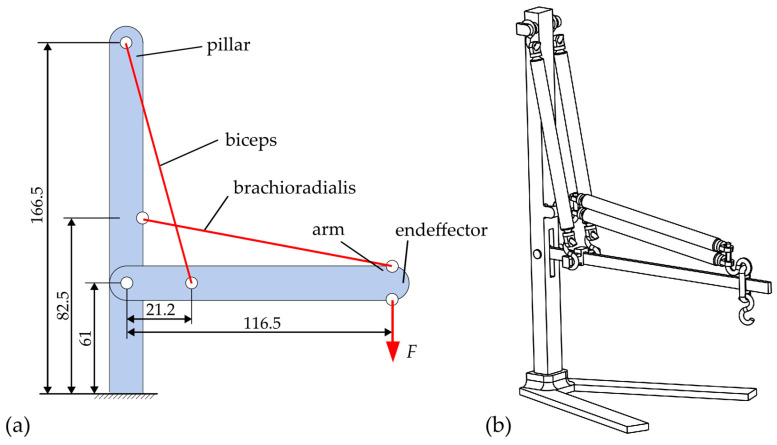
Schematic (**a**) and the final design (**b**) of the artificial muscle demonstrator.

**Figure 6 biomimetics-09-00236-f006:**
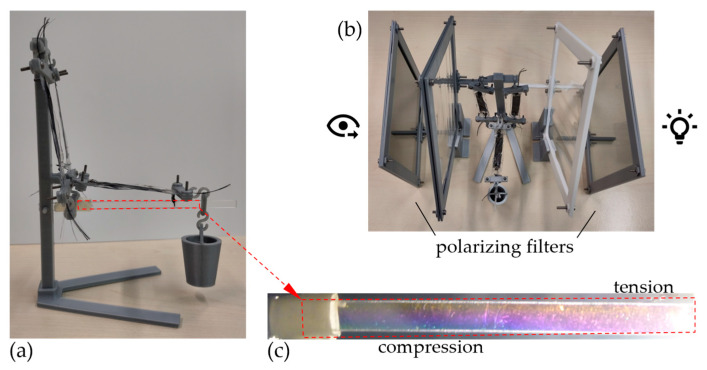
Overall view of the manufactured artificial muscle demonstrator (**a**), the photoelasticity setup (**b**), and the detailed view of the loaded PC arm (**c**).

**Figure 7 biomimetics-09-00236-f007:**
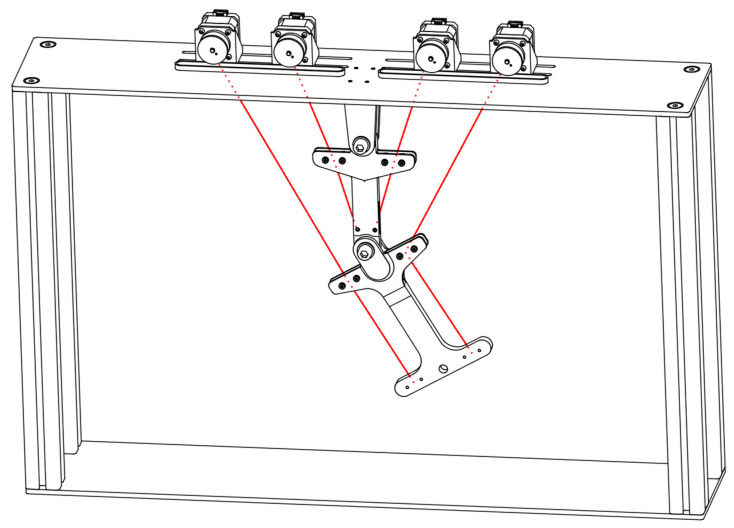
Final design of the rope pull demonstrator. The partially concealed rope pulls are shown in red.

**Figure 8 biomimetics-09-00236-f008:**
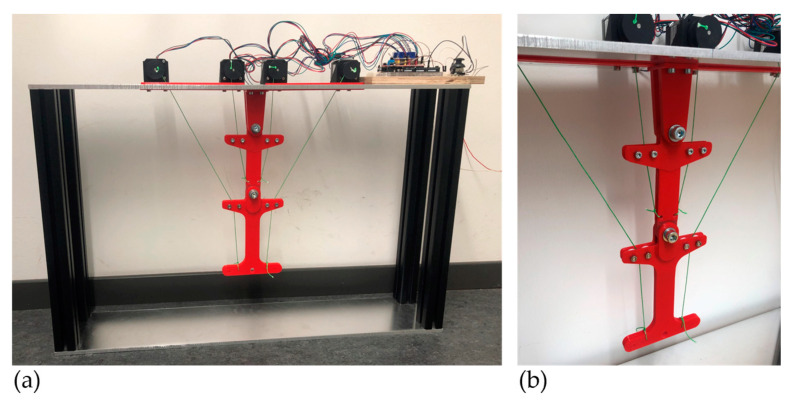
Overall view of the manufactured rope pull demonstrator (**a**) and detailed view of the robot arm (**b**).

**Table 1 biomimetics-09-00236-t001:** Advantages and disadvantages of artificial muscles and ropes as actuators for the technical implementation of the tension chording principle.

	Artificial Muscles	Ropes
**Advantages**	Close to the model of biological muscles; High compliance suitable for pretension and passive tension chording; No additional drive necessary on site; Direct attachment from segment to segment possible; Low moving masses.	Well-developed, technically already widespread; therefore, industrial experience is available; Easily commercially available in large quantities; Freedom in terms of rope materials, rope guidance, and motor positioning; Good scalability; High transferable loads; Low moving masses; Fast and precise control.
**Disadvantages**	In research stage, not yet widely used in industry; Rarely commercially available in larger quantities; Unfortunate dynamic behavior; Precise control is difficult; Partly, rather untypical aggregates such as high-voltage source, pneumatics, etc., are necessary; Poorly scalable, thus transferable loads and forces are limited.	Additional drives are necessary; Partly, more components such as deflection devices and cable guides are necessary; Additional joint forces generate bending, especially with non-redundant actuators.

## Data Availability

For insights into in-depth data, e.g., design documentation data, etc., please contact the corresponding author.

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
