# Peer review of "Assessment of the Suitability of Selected Linear Actuators for the Implementation of the Load-Adaptive Biological Principle of Redundant Motion Generation"

_biomimetics, 2024, doi:10.3390/biomimetics9040236_

Round 1
Reviewer 1 Report (New Reviewer)
Comments and Suggestions for Authors
The paper raises an interesting issue of application of the actuator redundancy for more efficient control of loads in artificial hand-like systems (manipulators), while only an initial step in this direction was demonstrated. The used biomimetic approach is within the scopes of the journal. The main part of this paper is a literature review and critical analysis in order to select appropriate technical solution. The review is supplemented by a brief description of two mechanical models not incorporating the redundancy principle. Although the research is rather preliminary, the preformed analyses and demonstrated technical solutions can be of interest for readers, and the paper potentially can be acceptable. The Authors should critically revise their paper in view of the following comments:
1. The paper is interesting, but, in its present form, it is hard to read, because main results are hidden in a large amount of text. The paper is overloaded by repetitions and can be considerably shortened by the expanse of their reduction. For instance, multiple sentences like “As mentioned above, ropes are passive elements and require a drive, e.g. a motor” are excessive. The literature review is bloated, tables can be beneficial to present the features of different solution in a condense manner. There are too many discussions through the text; improper variants can be less discussed, as well as obvious facts.
2. Section 1.1 can be shorten be excluding repetitions, because some of points are discussed repeatedly before and after consideration of Fig. 1.
3. References “upper part” or “middle part” of figure, please, replace by Figure 1(a) and so on, and update figures correspondingly.
4. The drive on the upper segment in Fig. 4(a) seems to be shown incorrect, because it will not move the segment in the shown configuration.
Author Response
See pdf

Reviewer 2 Report (New Reviewer)
Comments and Suggestions for Authors
This paper is well-written, and it is suitable for publication in the present form. If a revision is asked by the editor, the following point should be considered:
This paper considers a linear actuator, however for the practical applications, a variable stiffness actuator has to be considered, see for example,
Kozakiewicz, B and Winiarski, T. SPRING BASED ON FLAT PERMANENT MAGNETS: DESIGN, ANALYSIS AND USE IN VARIABLE STIFFNESS ACTUATOR, FACTA UNIVERSITATIS-SERIES MECHANICAL ENGINEERING 21 (1) , pp.101-120, 2023
Author Response
see pdf

Reviewer 3 Report (New Reviewer)
Comments and Suggestions for Authors
This article aims to a comparison of the performances of artifical muscles and ropes in actuators for robotics.
In the first part a review of sufficient literature is carried out, but at least some relevant pictures, graphs and tables should be shown, taken from the cited references.
In the second part, the building of two demostrators is shown, with different geometries, one using a particular artificial muscle (dielectric elastomer) and one using a kind of HM-HT rope. Anyway, apart from a qualitative photoelasticity measurement of stress distribution, carried out only in the arm of the artifical muscle demonstrator, no experimental results are presented to support the statements reported in the discussion and conclusion sections, which therefore appear too generic and qualitative.
On the other hand, the authors themselves admit in the discussion section, and say again in the conclusions, that "a number of aspects have not yet been investigated in this work, including a more detailed study of the dynamic behavior and control aspects".
Therefore, in my opinion this work is not ready for publication at this stage, it could be only once those investigations are performed and experimental results are presented.
Author Response
see pdf

Reviewer 4 Report (New Reviewer)
Comments and Suggestions for Authors
The manuscript analyses different advantages and disadvantages of the technical linear actuators considering that ropes are currently the best way to imitate the load-adaptive behaviour of the biological model in technology.
The manuscript is organized with tables and diagrams. The following concerns must be considered before it can be accepted for publication.
1. The meaning of the paper should be better pointed out.
2. English should be checked.
Comments on the Quality of English LanguageMinor editing of English language required
Author Response
see pdf

Reviewer 5 Report (New Reviewer)
Comments and Suggestions for Authors
The paper has an interesting theme and the problem addressed is very complex. I think that the work can be improved by the following additions:
1. the main difference between the osteo joint muscular system and the proposed technical approach consists in the fact that the biological system is fully elastic and the technical one has limited elasticity. This difference is important and should be highlighted.
2. the elements of the muscular system are functionally differentiated, the fibers, fascicles, groups and muscle chains work synergistically and antagonistically (during the flexion movement - for example, the extensors also act to "dosage" the movement and to brake it). I therefore consider that it is not a matter of redundancy in the engineering sense. Moreover, redundancy technically translates into a number of equations equal to the number of degrees of mobility and nothing is mentioned about this in the article.
3. the experimental component provides only flat cases, which greatly simplifies the problem. It should at least be remembered what happens in space cases.
Author Response
see pdf

Round 2
Reviewer 1 Report (New Reviewer)
Comments and Suggestions for Authors
My comments have been addressed.
Reviewer 3 Report (New Reviewer)
Comments and Suggestions for Authors
I have read the authors’ response, and I thank them for it.
Anyway, on the basis of their response, I am still not convinced about their paper.
Actually, as I already said in my first report, the first part of it could be a publishable review and I thank them for adding the table summarizing some of the findings of the cited literature.
On the contrary, in my opinion, the second part still does not conclude anything relevant, as there are no experimental results.
The only conclusion is that they were able to build the demonstrators, but at this stage they cannot say if they are really suitable for applications and if the materials/techniques used, and the demonstrators themselves, have better or worse performances with respect to similar devices. The only thing they can say is what the demonstrators “should be” according to literature. Again, my opinion is that this part could be publishable only once some experimental results are available, from which some quantitative conclusions can be drawn.
On the other hand, the authors declare in their response that other four reviewers agreed with the general approach of the article. Therefore, since it looks like I was the only one to recommend rejection of the article, finally I defer to the opinion of the majority of the reviewers and I do not oppose its publication.
This manuscript is a resubmission of an earlier submission. The following is a list of the peer review reports and author responses from that submission.
Round 1
Reviewer 1 Report
Comments and Suggestions for Authors
The result of the revision deepens the impression that the background of the manuscript is purely technical, and the biological background itself for the justification of the approach as only biological inspiration instead of bionics (as a more appropriate term for the vague term biomimetics) was not really understood. A clear definition of the principle derived from the observation of nature is missing. The standard literature used does not provide a sufficient basis for the specific interpretation with regard to the technical task; a single current publication on the form-function adaptation of bone tissue in the specific situation of implant environment does not provide a reliable basis for conclusions on generalisable principles.
There is also room for improvement in the engineering part of the text. If the dynamics are disregarded and the focus is limited to statics (which needs to be explained for a mechanism), there is no clear distinction between load and stress. Load is minimised by adapting the macromorphology of the musculoskeletal system, stress is optimised (not exclusively minimised) by adapting the micromorphology - an analogous technical approach is not visible.
In my opinion, the article does not represent a systematic bionic approach; bioinspiration has existed since the beginning of the Anthropocene and can be used without constraint as an introduction to a purely technical article, which can then also be subjected to a purely technical expert assessment. The authors' technical expertise, which is not denied, could then also be developed in a more targeted manner, and the topic addressed is both interesting and topical.
In its present form, the article is not convincing from either a biological or a technical perspective. In particular, I do not see a truly scientifically based red line from biological modelling to technical implementationand I do not see any clearly formulated solution as plausible evidence for the claim „great potential for minimizing resource and energy requirements“ (first sentence in abstract).
Reviewer 2 Report
Comments and Suggestions for Authors
My original concerns were two folds 1) the focus was blurred between literature review and demonstrator designs, 2) if latter, there was not sufficient quantitative design details, justification and assessment. Just because the Authors have modified the words “literature review” to e.g., “purposeful literature summary”, it doesn’t change the fact that there is substantial amount of literature information in the draft while only partial content is actually directly linked to the final design demos. Furthermore, to my second point, simply saying that the two separate exampled actuators were chosen because they were practically available in the research group i.e., as “knowledge-generating” attempts is just not rigorous enough for a scientific publication, despite the face it is true. Detailed scientific justifications (stemmed from the literature summary) should be added. In addition, I want to reiterate the complete lack of design details and justifications, e.g., how exactly those dimensions came about at the first place e.g., in Figure 5 and 7 and what kind of load do you expect from those designs? Even these are individual “proof of concepts” demos, somebody somewhere must have gone through the design analysis in order to arrive at a “final design” in Figure 7? What are the key considerations in these dimensions of the design and what have been the process and why? All these would be needed to justify rigorous research which we should expect in such a paper. Since the Authors haven’t addressed these fundamental concerns in the revised manuscript, unfortunately, with sincere regret, I don’t feel the revised manuscript in its current form is acceptable for publication.